A new limestone-dwelling species of Micryletta (Amphibia: Anura: Microhylidae) from northern Vietnam

http://orcid.org/0000-0002-7576-2283 Poyarkov Nikolay A. 1 2 n.poyarkov@gmail.com
Nguyen Tan Van 3 4
http://orcid.org/0000-0001-9943-8843 Duong Tang Van 1 5
Gorin Vladislav A. 1
Yang Jian-Huan 6
1 Faculty of Biology, Department of Vertebrate Zoology, Moscow State University , Moscow , Russia
2 Laboratory of Tropical Ecology, Joint Russian-Vietnamese Tropical Research and Technological Center , Hanoi , Vietnam
3 Save Vietnam’s Wildlife , Ninh Binh , Vietnam
4 Institute of Research and Development, Duy Tan University , Da Nang City , Vietnam
5 Vietnam National Museum of Nature, Vietnam Academy of Science and Technology , Hanoi , Vietnam
6 Kadoorie Conservation China, Kadoorie Farm and Botanic Garden , Hong Kong , China
Pie Marcio
Electronic publication date: 2018 Oct 4
Publication date: 2018
Volume: 6
Electronic Location ID: e5771
Received 2018 Aug 22; Accepted 2018 Sep 17
Copyright: © 2018 Poyarkov et al.
Copyright year: 2018
Copyright holder: Poyarkov et al.
License: This is an open access article distributed under the terms of the Creative Commons Attribution License, which permits unrestricted use, distribution, reproduction and adaptation in any medium and for any purpose provided that it is properly attributed. For attribution, the original author(s), title, publication source (PeerJ) and either DOI or URL of the article must be cited.
License URL: https://creativecommons.org/licenses/by/4.0/

Keywords: Phylogeny, Taxonomy, Indochina, Morphology, Red river, Micryletta nigromaculata sp. nov, Endemism, Karst, mtDNA, Biodiversity

Funding: Joint Russian-Vietnamese Tropical Research and Technological Center, Hanoi, Vietnam Russian Science Foundation 14-50-00029 Fieldwork was supported by the Joint Russian-Vietnamese Tropical Research and Technological Center, Hanoi, Vietnam. Molecular experiments, phylogenetic analyses, specimen storage and examination were carried out with the financial support of Russian Science Foundation (RSF grant No. 14-50-00029). The funders had no role in study design, data collection and analysis, decision to publish, or preparation of the manuscript.

==============================
We report on a new species of the genus Micryletta from limestone karst areas in northern Vietnam, which is described on the basis of molecular and morphological evidence. Micryletta nigromaculata sp. nov. is restricted to narrow areas of subtropical forests covering karst massifs in Cat Ba National Park (Hai Phong Province) and Cuc Phuong National Park (Ninh Binh Province) at elevations of 90–150 m a.s.l. In the phylogenetic analyses, the new species is unambiguously positioned as a sister lineage to all remaining species of Micryletta. We also discuss genealogical relationships and taxonomic problems within the genus Micryletta, provide molecular evidence for the validity of M. erythropoda and discuss the taxonomic status of M. steinegeri. We suggest the new species should be considered as Endangered (B1ab(iii), EN) following the IUCN’s Red List categories. A discussion on herpetofaunal diversity and conservation in threatened limestone karst massifs in Southeast Asia is provided.

Introduction

Paddy frogs of the genus Micryletta Dubois (1987) are a little-known group of microhylids that occurs from southern China, Taiwan, Thailand, Indochina and Myanmar to Nicobar and the Andaman Islands (India), West Malaysia and Sumatra (Indonesia) (Frost, 2018). To date, three species are recognized within the genus: M. inornata (Boulenger, 1890) (type locality: Sumatra, Indonesia; distributed in Sumatra, Nicobar and the Andaman Islands, Peninsular Malaysia, Indochina and southern China), M. steinegeri (Boulenger, 1909) (distributed in southern Taiwan and China) and M. erythropoda (Tarkhnishvili, 1994) (distributed in lowlands of southern Vietnam) (AmphibiaWeb, 2018; Frost, 2018). All these species were initially described within the genus Microhyla Tschudi, 1838; however, Dubois (1987) erected a new genus Micryletta, distinguishing it from Microhyla on the basis of a suite of characters including: snout shorter than the eye and eye less prominent (vs. opposite condition in Microhyla); distinct tympanum (vs. hidden in Microhyla); first finger not reduced (vs. opposite condition in some species of Microhyla); digit tips not expanded into disks (vs. expanded in most species of Microhyla); and webbing totally absent in Micryletta (vs. always present in Microhyla) (Dubois, 1987; Bain & Nguyen, 2004).

Owing to morphological conservativeness, biodiversity of the genus Micryletta is insufficiently studied and its taxonomy was confusing. For instance, M. steinegeri, endemic to Taiwan, was synonymized with M. inornata due to the difficulty in distinction between these two species confused with M. inornata (Parker, 1928, 1934; Wang, Wu & Yu, 1989). Morphological study by Dubois (1987) supported the validity of M. steinegeri, which was followed by Fei et al. (2009), Fei, Ye & Jiang (2010) but rejected by Zhao & Adler (1993), while Matsui & Busack (1985) confirmed synonymy of Rana gracilipes Gressitt with M. steinegeri. Validity of a subspecies M. inornata lineata (Taylor, 1962) described from southern Thailand was not examined by latter studies. Finally, Microhyla erythropoda Tarkhnishvili, 1994 described from two specimens from southern Vietnam was assigned to the genus Micryletta by Orlov et al. (2002) and Poyarkov et al. (2014), but without providing details on taxonomy of this group.

Works on molecular phylogenetic relationships of the genus Micryletta are scarce. Van Der Meijden et al. (2007) as well as Pyron & Wiens (2011) confirmed the validity of the genus Micryletta and suggested that Micryletta is a sister taxon to the group composed of Microhyla and Glyphoglossus s.lato (including Caluella), though with low values of node support. Matsui et al. (2011) provided an extensive phylogeny of Asian Microhylinae on the basis of 12S rRNA and 16S rRNA mtDNA data, in their tree phylogenetic position of Micryletta within Microhylidae is not supported, though the data suggest paraphyly of M. inornata with respect to M. steinegeri. The recent phylogenomic work by Peloso et al. (2016) also unambiguously places the genus Micryletta as a sister taxon to the group composed of Microhyla and Glyphoglossus, while a more recent large-scale multilocus phylogeny by Tu et al. (2018) on the contrary places Micryletta as a sister lineage of the clade joining Uperodon, Phrynella, Metaphrynella and Kaloula. Thus, phylogenetic placement of Micryletta within Microhyinae is still unresolved, and species-level phylogeny of the genus is still absent.

During our recent fieldwork in northern Vietnam, in the limestone forests of Hai Phong and Ninh Binh provinces we encountered unusual microhylid specimens, which were tentatively identified as Micryletta sp. Consequent phylogenetic analysis of the 16S rRNA mtDNA gene revealed that these populations form a lineage sister to all other recognized species of the genus Micryletta. Closer morphological examination showed that the specimens from Hai Phong and Ninh Binh provinces are clearly distinguished from other known members of Micryletta by a combination of diagnostic morphological features. In the present paper, we provide an updated mtDNA-based genealogy of the genus Micryletta and describe a new species from northern Vietnam.

Materials and Methods

Nomenclatural acts

The electronic version of this article in portable document format will represent a published work according to the International Commission on Zoological Nomenclature (ICZN), and hence the new names contained in the electronic version are effectively published under that Code from the electronic edition alone (see Articles 8.5–8.6 of the Code). This published work and the nomenclatural acts it contains have been registered in ZooBank, the online registration system for the ICZN. The ZooBank LSIDs (Life Science Identifiers) can be resolved and the associated information can be viewed through any standard web browser by appending the LSID to the prefix http://zoobank.org/. The LSID for this publication is as follows: urn:lsid:zoobank.org:pub:32150A60-5D04-4116-9816-0ED3E457504A. The online version of this work is archived and available from the following digital repositories: PeerJ, PubMed Central and CLOCKSS.

Sample collection

Fieldwork was conducted from 10 to 22 October 2013 by Nikolay A. Poyarkov and Jian-Huan Yang in Cat Ba National Park (hereafter—N. P.), Hai Phong Province; and from 8 to 17 June 2017 by Tan Van Nguyen and Tan Nhat La in Cuc Phuong N. P., Ninh Binh Province of northern Vietnam. Surveyed localities are shown in Fig. 1. Geographic coordinates and elevation were obtained using a Garmin GPSMAP 60CSx (USA) and recorded in WGS84 datum. All specimens were preserved in 75% ethanol, and muscle tissues were preserved in 95% ethanol for genetic analysis; the holotype specimen was initially fixed in 4% formalin for 24 h and later preserved in 75% ethanol. Specimens and tissues were subsequently deposited in the zoological collections of the Zoological Museum of Lomonosov Moscow State University (ZMMU), Moscow, Russia, the Duy Tan University (DTU), Da Nang Province, Vietnam and the Museum of Biology, Sun Yat-sen University (SYS), Guangzhou, China. Comparative materials examined are stored in the herpetological collections of ZMMU and in the Zoological Institute of the Russian Academy of Sciences (ZISP) in St. Petersburg, Russia.

Figure 1 Distribution of the genus Micryletta and the new species.

(A) Map of Southeast Asia with approximate range of the genus Micryletta shown in red. Black circles indicate type localities of the currently recognized taxa within Micryletta. Yellow stars show distribution of Micryletta nigromaculata sp. nov. black dot in the center of icon indicates the type locality (Cat Ba Island). Black square indicates the inset shown in detail in B. (B) Map of northern Vietnam, showing distribution of Micryletta nigromaculata sp. nov. and the Red River basin; 1—Cat Ba National Park, Hai Phong Province (type locality); 2—Cuc Phuong National Park, Ninh Binh Province. Photo by Nikolay A. Poyarkov.

Specimens collection protocols and animal use were approved by the animal operations were approved by the Institutional Ethical Committee of Animal Experimentation of Sun Yat-sen University (certificate number 2005DKA21403-JK issued to Ying-Yong Wang and Jian-Huan Yang). Fieldwork, including collection of animals in the field, was authorized by the Department of Forestry, Ministry of Agriculture and Rural Development of Vietnam (permit number 1461/TCLN-BTTN, issued September 23, 2013).

Laboratory methods

For the molecular phylogenetic analyses, we extracted total genomic DNA from ethanol-preserved femoral muscle tissue using standard phenol-chloroform-proteinase K extraction procedures with consequent isopropanol precipitation, for a final concentration of about one mg/ml (protocols followed Hillis, Moritz & Mable, 1996 and Sambrook & David, 2001). We visualized the isolated total genomic DNA in agarose electrophoresis in presence of ethidium bromide. We measured the concentration of total DNA in one μl using NanoDrop 2000 (Thermo Scientific, Waltham, MA, USA), and consequently adjusted to ca. 100 ng DNA/μl.

We amplified mtDNA fragments covering partial sequences 16S rRNA mtDNA gene to obtain a 947 bp-length continuous fragment of mtDNA. 16S rRNA gene was widely applied in biodiversity surveys in amphibians (Vences et al., 2005a, 2005b; Vieites et al., 2009), and has been used in the most of recent phylogenetic studies on Microhylinae (Matsui et al., 2011; Peloso et al., 2016; Tu et al., 2018). We performed DNA amplification in 20 μl reactions using ca. 50 ng genomic DNA, 10 nmol of each primer, 15 nMol of each dNTP, 50 nMol additional MgCl2, Taq PCR buffer (10 mM Tris-HCl, pH 8.3, 50 mM KCl, 1.1 mM MgCl2 and 0.01% gelatine) and 1 unit of Taq DNA polymerase. Primers used in PCR and sequencing include: L-2188 (AAAGTGGGCCTAAAAGCAGCCA), 16sL1 (CTGACCGTGCAAAGGTAGCGTAATCACT) and 16H-1 (CTCCGGTCTGAACTCAGATCACGTAGG) (Matsui et al., 2006; Hedges, 1994). The PCR conditions included an initial denaturation step of 5 min at 94 °C and 43 cycles of denaturation for 1 min at 94 °C, primer annealing for 1 min with TouchDown program from 65 to 55 °C reducing 1 °C every cycle, and extension for 1 min at 72 °C, and final extension step for 5 min at 72 °C.

PCR products were loaded onto 1.5% agarose gels in presence of ethidium bromide and visualized in agarose electrophoresis. When distinct bands were produced, we purified PCR products using two μl of a 1:4 dilution of ExoSapIt (Amersham) per five μl of PCR product prior to cycle sequencing. A 10 μl sequencing reaction included two μl of template, 2.5 μl of sequencing buffer, 0.8 μl of 10 pmol primer, 0.4 μl of BigDye Terminator version 3.1 Sequencing Standard (Applied Biosystems, Foster City, CA, USA) and 4.2 μl of water. The cyclesequencing used 35 cycles of 10 s at 96 °C, 10 s at 50 °C and 4 min at 60 °C. We purified the cyclesequencing products by ethanol precipitation. We carried out sequence data collection and visualization on an ABI 3730xl Automated Sequencer (Applied Biosystems, Foster City, CA, USA). The obtained sequences are deposited in GenBank under the accession numbers MH756146–MH756156 and MH879840–MH879845 (Table S1).

Phylogenetic analyses

To reconstruct the matrilineal genealogy, we used all 16S rRNA sequences for Micryletta available in GenBank and our newly obtained sequences of Micryletta sp. and sympatric populations of Micryletta cf. inornata (see Table S1). We also added sequences of representatives of all currently recognized Microhylinae genera and a sequence of Chaperina fusca (Chaperininae). In total, we obtained data for 16S rRNA for 43 specimens, which included six sequences of Micryletta sp. from Cat Ba Island, three sequences of Micryletta sp. from Cuc Phuong N. P., three sequences of Micryletta cf. inornata from Cat Ba Island, three sequences of Micryletta cf. inornata from Cuc Phuong N.P., 11 sequences of all other species of Micryletta from Thailand, Laos, Vietnam and Taiwan, including topotype specimens of M. erythropoda (Ma Da, Dong Nai, Vietnam) and M. steinegeri (Taiwan), 16 outgroup sequences of other Microhylinae representatives and of Chaperina fusca, and a sequence of Kalophrynus interlineatus (Blyth) (Kalophryninae), which was used to root the tree (data summarized in Table S1).

We initially aligned nucleotide sequences using ClustalX 1.81 (Thompson et al., 1997) with default parameters, and then optimized them manually in BioEdit 7.0.5.2 (Hall, 1999) and MEGA 6.0 (Tamura et al., 2013). We used MODELTEST v.3.06 (Posada & Crandall, 1998) to estimate the optimal evolutionary models to be used for the data set analysis. The best-fitting model for the 16S rRNA gene fragment was the GTR+G model of DNA evolution as suggested by the Akaike Information Criterion. We determined mean uncorrected genetic distances (p-distances) between sequences with MEGA 6.0.

We inferred the matrilineal genealogy using Bayesian inference (BI) and Maximum Likelihood (ML) approaches. We conducted BI in MrBayes 3.1.2 (Ronquist & Huelsenbeck, 2003); Metropolis-coupled Markov chain Monte Carlo (MCMCMC) analyses were run with one cold chain and three heated chains for one million generations and sampled every 100 generations. We performed five independent MCMCMC runs and the initial 2,500 trees were discarded as burn-in. We assessed confidence in tree topology by the frequency of nodal resolution (posterior probability; BI PP) (Huelsenbeck & Ronquist, 2001). We conducted ML analyses using the RAxML web server (http://embnet.vital-it.ch/raxml-bb/, Stamatakis, Hoover & Rougemont, 2008); it was used to search ML trees using the gamma model of rate heterogeneity option. We assessed nodal confidence by non-parametric bootstrapping (ML BS) with 1,000 pseudoreplicates (Felsenstein, 1985).

In both datasets, we regarded tree nodes with ML BS values 75% or greater and BI PP values over 0.95 to be sufficiently resolved a priori. ML BS values between 75% and 50% and BI PP values between 0.95 and 0.90 were regarded as tendencies. Lower values were considered to indicate unresolved nodes (Huelsenbeck & Hillis, 1993).

Morphological description

Specimens of Micryletta sp. were photographed in life and after preservation; specimens were euthanized by 20% solution of benzocaine. Measurements were taken using a digital caliper under a light dissecting microscope to the nearest 0.01 mm, subsequently rounded to 0.1 mm. The morphometrics of adults and character terminology follow Poyarkov et al. (2014): (1) snout–vent length (SVL; measured from the tip of the snout to cloaca); (2) head length (HL; measured from the tip of snout to hind border of jaw angle); (3) snout length (SL; measured from the anterior corner of eye to the tip of snout); (4) eye length (EL; measured as the distance between anterior and posterior corners of the eye); (5) nostril–eye length (measured as the distance between the anterior corner of the eye and the nostril center); (6) head width (HW; measured as the maximum width of head on the level of mouth angles in ventral view); (7) internarial distance (IND; measured as the distance between the central points of nostrils); (8) interorbital distance (IOD; measured as the shortest distance between the medial edges of eyeballs in dorsal view); (9) upper eyelid width (UEW; measured as the maximum distance between the medial edge of eyeball and the lateral edge of upper eyelid); (10) Tympanum length, measured as the horizontal tympanum diameter; (11) forelimb length (FLL; measured as the length of straightened forelimb to the tip of third finger); (12) lower arm and hand length (LAL; measured as the distance between elbow and the tip of third finger); (13) hand length (HAL; measured as the distance between the proximal end of outer palmar (metacarpal) tubercle and the tip of third finger); (14) first finger length (1FL, measured as the distance between the tip and the distal end of inner palmar tubercle); (15) inner palmar tubercle length (IPTL; measured as the maximum distance between proximal and distal ends of inner palmar tubercle); (16) outer palmar tubercle length (OPTL; measured as the maximum diameter of outer palmar tubercle); (17) third finger disk diameter (3FDD); (18) hindlimb length (HLL; measured as the length of straightened hindlimb from groin to the tip of fourth toe); (19) tibia length (TL; measured as the distance between the knee and tibiotarsal articulation); (20) foot length (FL; measured as the distance between the distal end of tibia and the tip of fourth toe); (21) inner metatarsal tubercle length (IMTL; measured as the maximum length of inner metatarsal tubercle); (22) first toe length (1TOEL), measured as the distance between the distal end of inner metatarsal tubercle and the tip of first toe; (23) fourth toe disk diameter (4TDD). Additionally for holotype description we took the following measurements: (24–26) second to fourth finger lengths (2–3FL-O, 4FL-I; for outer side (O) of the second and third, inner side (I) of the fourth, measured as the distance between the tip and the junction of the neighboring finger); (27–30) second to fifth toe lengths (measured as the outer lengths for toes II–IV, as the inner length for toe V; 2–5TOEL); (31) nostril–snout length (N–SN), measured as the distance between the middle of nostril and snout tip. Terminology for describing eye coloration in living individuals is in accordance with Glaw & Vences (1997); subarticular tubercle formulas follow those of Savage (1975). All measurements were taken on the right side of the examined specimen. Sex was determined by gonadal inspection following dissection.

We compared morphological characters of the new species with other members of the genus and comparative data obtained from the literature: Micryletta inornata (Boulenger) (Boulenger, 1890; Taylor, 1962; Bain & Nguyen, 2004), M. steinegeri Boulenger (Boulenger, 1909; Wang, Wu & Yu, 1989; Fei et al., 2009; Fei, Ye & Jiang, 2010) and M. erythropoda (Tarkhnishvili) (Tarkhnishvili, 1994).

For preparing maps free Shuttle Radar Topography Mission Digital Elevation Datasets (accessed from ftp://e0srp01u.ecs.nasa.gov/srtm) and free country level GIS data downloaded from DIVA-GIS portal (http://www.diva-gis.org/gdata) were processed using the Global Mapper v. 10.0 software (Global Mapper, 2009).

Results

Phylogenetic relationships

In the final alignment of 16S rRNA gene, of 947 sites 554 were conserved and 346 sites were variable, of which 250 were found to be parsimony-informative. The transition–transversion bias (R) was estimated as 2.62. Nucleotide frequencies were A = 33.81%, T = 23.23%, C = 23.88% and G = 19.04% (data for ingroup only).

The studied 16S rRNA fragment was unable to fully resolve the genealogical relationships within Microhylinae (see Fig. 2). The genus Micryletta is suggested as a sister lineage to the clade joining Kaloula, Phrynella, Metaphrynella and Uperodon, though with moderate levels of node support (0.94/78, hereafter node support values are given for BI PP/ML BS, respectively). According to the results of phylogenetic analyses, the newly discovered populations of Micryletta sp. from northern Vietnam form a well-supported clade (1.0/100) markedly distinct from all other examined Microhylinae representatives. The Micryletta sp. clade is reconstructed as a sister lineage to all other Micryletta specimens (1.0/97), the latter also forming a clade (1.0/95) (Fig. 2). Genealogical relationships within this group suggest that specimens from the type locality of M. erythropoda (Ma Da Nature Reserve, Dong Nai Province, Vietnam) are clustered with a sample of Micryletta sp. from Ranong Province, Thailand (1.0/100) (the latter sample was previously assigned to M. inornata lineata in Matsui et al., 2011). This clade is a sister lineage to the group joining all remaining specimens of Micryletta inornata and M. steinegeri. Evolutionary relationships within the latter group are essentially unresolved with the Taiwanese population of M. steinegeri being nested within the radiation of mainland populations of M. inornata (see Fig. 2). Populations of M. cf. inornata sympatric with the new species in Cat Ba and Cuc Phuong national parks form a weakly supported clade (M. cf. inornata B; 0.77/91).

Figure 2 Phylogenetic BI tree of Micryletta reconstructed on the base of 947 bp of 16S rRNA gene.

Values on the branches correspond to BI PP/ML BS, respectively. For specimen, locality and GenBank accession number information see Table 1. Photos by Nikolay A. Poyarkov (Micryletta nigromaculata sp. nov. M. erythropoda, M. cf. inornata) and Chung-Wei You (M. steinegeri).

Sequence variation

The uncorrected p-distances for the 16S rRNA gene fragment are shown in the Table 1. The interspecific distances within Micryletta varied from p = 1.4% (between M. steinegeri and M. cf. inornata B) to p = 7.7% (between M. erythropoda and Micryletta sp. from northern Vietnam). Intraspecific distances ranged from p = 0.7% in Micryletta sp. from northern Vietnam (divergence between Cat Ba and Cuc Phuong populations) to p = 2.2% in M. inornata (but only p = 1.3% upon the exclusion of a notably divergent KUHE 35133 sample from Laos, see Table 1). The newly discovered population of Micryletta sp. was clearly divergent from all other known species of Micryletta and other examined microhylids.

Table 1 Uncorrected p-distance (percentage) between 16S rRNA sequences of Micryletta and other microhylids included in the phylogenetic analyses (below the diagonal), and standard error estimates (above the diagonal).

	Taxon	1	2	3	4	5	6	7	8	9	10	11	12	13	14	
	Ingroup: Micryletta		
1	M. nigromaculata sp. nov.	0.7	1.2	1.1	1.2	1.1	1.4	1.4	1.4	1.3	1.4	1.6	1.3	1.7	1.5	
2	M. cf. inornata A	7.2	2.2	0.7	0.6	1.0	1.5	1.4	1.4	1.4	1.4	1.6	1.4	1.8	1.6	
3	M. cf. inornata B	5.9	2.5	0.1	0.5	0.9	1.5	1.4	1.4	1.4	1.4	1.7	1.4	1.8	1.5	
4	M. steinegeri	5.9	2.8	1.4	—	0.9	1.5	1.4	1.4	1.4	1.4	1.6	1.4	1.8	1.6	
5	M. erythropoda	7.7	5.9	4.6	5.5	1.7	1.5	1.4	1.4	1.3	1.4	1.6	1.3	1.7	1.6	
	Outgroup		
6	Microhyla I	10.5	11.4	11.4	10.8	12.1	—	1.4	1.5	1.3	1.6	1.8	1.7	2.0	1.4	
7	Microhyla II	12.4	12.7	12.5	12.0	13.4	12.0	7.3	1.3	1.4	1.6	1.6	1.5	1.7	1.5	
8	Glyphoglossus	12.0	12.5	12.1	11.8	13.6	11.9	12.4	8.6	1.3	1.3	1.6	1.5	1.6	1.5	
9	Kaloula	12.3	12.6	12.2	12.2	12.8	11.6	14.2	14.1	6.8	1.1	1.4	1.3	1.7	1.4	
10	Uperodon	13.6	13.5	12.9	13.3	13.3	13.3	14.7	12.1	10.0	—	1.6	1.3	1.9	1.6	
11	Phrynella	14.7	13.8	13.5	12.9	14.4	15.0	14.9	15.5	12.3	12.9	—	1.2	2.0	1.6	
12	Metaphrynella	14.2	13.1	13.0	12.5	14.7	14.4	14.9	15.8	11.7	11.9	8.2	7.1	1.9	1.5	
13	Chaperina	16.1	17.7	17.2	16.9	17.0	18.1	18.3	16.0	19.6	21.5	19.6	20.6	—	1.9	
14	Kalophrynus	12.9	14.7	14.0	13.9	15.2	13.6	13.3	15.2	15.6	16.7	16.7	15.6	19.4	—	
Note:

The ingroup mean uncorrected p-distances are shown on the diagonal and shaded with gray.

Taxonomic Account

The newly-discovered populations of microhylids from Cat Ba and Cuc Phuong are clustered with the genus Micryletta, forming a divergent lineage sister to all other representatives of the genus examined. Due to both morphological (see below) and molecular differences of the newly-collected specimens to all currently recognized species in the genus, herein we describe it as a new species of Micryletta.

Micryletta nigromaculata sp. nov.

(Figs. 3–6; Table 2)

Holotype. ZMMU A5934, adult male (field ID NAP-06531), collected by Nikolay A. Poyarkov on October 15, 2013 from the limestone evergreen forest (20.8123°N, 106.9988°E, at an elevation of 90 m a.s.l.), Cat Ba National Park, Hai Phong Province, northern Vietnam.

Paratypes. ZMMU A5935–A5948 (14 adult males, field IDs NAP-03343–03348; NAP-03576–03579; NAP-03589–03590; NAP-08445-08446), with collection information same as for the holotype; SYS a007400 (field ID NAP-08444), adult male with collection information same as for the holotype.

Referred materials. DTU 301–302 (two males; field IDs CP.2018.18, CP.2018.31), DTU 303–305 (three gravid females, field IDs CP.2018.19, CP.2018.20, CP.2018.52) collected by Tan Van Nguyen and Tan Nhat La on June 3, 2018 in the secondary forest on limestone (20.2594°N, 105.6928°E, at elevation of 150 m a.s.l.), Cuc Phuong National Park, Ninh Binh Province, northern Vietnam.

Diagnosis. The new species is assigned to the genus Micryletta by the following combination of morphological features: small body size; vomerine teeth absent; tympanum small, rounded, externally visible; very prominent subarticular tubercles on fingers and toes; three well-developed metacarpal tubercles; distinct supernumerary palmar and metatarsal tubercles posterior to base of digits; first finger not reduced; digit tips expanded to very small disks and webbing on fingers and toes totally absent (Dubois, 1987; Fei et al., 2009). Micryletta nigromaculata sp. nov. is distinguished from all of its congeners by a combination of the following morphological characters: body size small (SVL 18.5–23.0 mm in males, 24.2–25.9 mm in females); body habitus moderately slender; head wider than long; snout obtusely rounded in profile; EL equal to or shorter than SL; IOD two times wider than UEW; tibiotarsal articulation of adpressed limb reaching the level of eye center; dorsal surface slightly granular with small round flattened tubercles; supratympanic fold present, thick, glandular; outer metatarsal tubercle absent; dorsum coloration brown to reddish-brown; dorsum with dark-brown irregular hourglass-shaped pattern edged with orange; body flanks brown with dark-brown to black patches or spots edged with white, a large black blotch in inguinal area on each side; lateral sides of head immaculate reddish brown lacking white patches; venter whitish with indistinct gray pattern; and throat in males whitish with light-gray marbling.

Description of holotype: Adult male, small-sized specimen in a good state of preservation; body habitus moderately slender, body elongated oval-shaped (Figs. 3 and 4); head wider than long (HL/HW ratio 84.4%); snout short (SL/SVL ratio 12.1%), rounded in dorsal view (Fig. 3A) and bluntly rounded in profile, slightly projecting beyond lower jaw (Fig. 3C); eyes comparatively large (EL/SVL ratio 12.6%), slightly protuberant in dorsal and lateral views, slightly longer than snout (EL/SL 104.0%) and shorter than the interorbital distance (EL/IOD 86.4%). Top of head flat; canthus rostralis distinct, rounded; loreal region almost vertical, noticeably concave; nostril round, lateral, located closer to the tip of snout than to eye (N-EL/SVL ratio 8.5%; N–SN/ N-EL ratio 79.0%) (Fig. 3C); interorbital distance wider than internarial distance (IND/IOD ratio 66.7%), about two times wider than upper eyelid (UEW/IOD ratio 54.5%). Pineal spot absent; tympanum small (TYD/SVL ratio 5.1%), round, relatively indistinct with tympanic rim not elevated above the tympanal area; supratympanic fold thick, rounded, glandular, gently curving from posterior corner of eye towards axilla. Choanae elongated and oval-shaped, widely spaced; upper jaw edentate; vomerine teeth absent; tongue without papillae, roundly spatulate, lacking posterior notch and free behind for 3/4 of its length.

Figure 3 Holotype of Micryletta nigromaculata sp. nov. (ZMMU A5934), male, in life.

(A) Dorsal view; (B) ventral view; (C) lateral view of head; (D) volar view of left hand; (E) plantar view of right foot. Photos by Nikolay A. Poyarkov.

Figure 4 Holotype of Micryletta nigromaculata sp. nov. (ZMMU A5934), male, in situ in dorsolateral view.

Photo by Nikolay A. Poyarkov.

Forelimbs short and slender (FLL/SVL ratio 72.0%); lower arm comparatively long and slender (LAL/SVL ratio 54.1%), hand less than half the length of the forelimb (HAL/FLL ratio 42.7%). Fingers slender (Figs. 3D and 5A), completely free of webbing, slightly dorso-ventrally flattened, lacking lateral skin fringes; the first finger well-developed, slightly shorter than the second finger (1FL/2FL ratio 74.8%); relative finger lengths: I < IV < II < III; tips of all finger rounded, not expanded to disks; subarticular tubercles on fingers rounded and very prominent, subarticular tubercle formula: 1, 1, 2, 2; nuptial pad absent; three palmar (metacarpal) tubercles: inner metacarpal tubercle distinct, rounded and flat (IPTL/SVL ratio 2.4%); outer metacarpal tubercle elongated, reniform, located on the outer proximal edge of the palm (OPTL/SVL ratio 3.3%); medial metacarpal tubercle large, rounded and prominent, twice the diameter of the inner metacarpal tubercle, located closer to the outer metacarpal tubercle; three rounded and prominent supernumerary palmar tubercles each at the base of fingers II–IV about the same size as inner metacarpal tubercle, a small rounded supernumerary palmar tubercle between medial metacarpal tubercle and the tubercle at the base of finger III, much smaller than metacarpal tubercles.

Figure 5 Morphological details of the male paratype of Micryletta nigromaculata sp. nov. (ZMMU A5945) in preservative.

(A) volar view of the right hand; (B) plantar view of the right foot. Scale bar equals three mm. Drawings by Valentina D. Kretova.

Hindlimbs slender and comparatively long (HLL/SVL ratio 152.9%), more than two times the length of the forelimb (FLL/HLL 47.1%); tibia long and slender (TL/SVL 51.4%), around one-third of hindlimb length (TL/HLL 33.6%); heels meet when hindlimbs located at right angles to the body, tibiotarsal articulation of adpressed limb reaching the level of eye center; foot slightly longer than tibia length (FL/TL 105.3%). Relative toe lengths: I < V < II < III < IV; tarsus smooth, inner tarsal fold absent; tips of all toes rounded, weakly dilated into small disks, slightly wider than those of fingers (3FDD/4TDD ratio 76.1%); toes completely free of webbing (Figs. 3E and 5B); subarticular tubercles on toes round and prominent, subarticular tubercle formula: 1, 1, 2, 3, 2; metatarsal tubercle single: inner metatarsal tubercle oval-shaped, prominent, much shorter than the half of first toe (IMTL/1TOEL ratio 36.8%); outer metatarsal and supernumerary metatarsal tubercles absent.

Skin texture and skin glands: Dorsal surface of head and body slightly granular with few small round low tubercles and granules evenly scattered being more prominent in the posterior part of dorsum, dorsal surfaces of forelimbs smooth, dorsal surfaces of hindlimbs covered by irregularly scattered flat tubercles and pustules; flanks of body and lateral sides of head smooth, with small granules present only in axillary region; upper eyelid without superciliary spines; supratympanic fold thick and glandular; ventral side of body and limbs smooth. Cloacal opening unmodified, directed posteriorly, at upper level of thighs.

Coloration in life: Dorsum coloration in life reddish-brown (Figs. 3A and 4); dorsal surfaces of forelimbs light brownish-orange on upper arms, reddish-brown on lower arms, dorsal surfaces of hindlimbs slightly darker and tan-brownish to caramel-brown in coloration; dorsal surfaces with distinct dark pattern: forehead and snout lighter; an distinct light-brownish interorbital bar runs transversally across the head between the medial parts of upper eyelids; interorbital bar forms a very distinct broad V-shaped figure across the head running posteriorly forming irregular hourglass-shaped dark-brown pattern; two smaller blotches in scapular region; dark pattern on dorsum edged with thin light-brown to orange line; head laterally dark red-brown, supratympanic fold black ventrally, edged with light cream-beige thin line dorsally, which continues to upper eyelid and canthus rostralis (Fig. 4); flanks with white speckling and characteristic large black patches edged with thin white lines; larger black blotches located at axillary and groin areas, the latter reaching the sacral area; fingers and toes dorsally beige with indistinct brownish mottling, venter whitish, with indistinct light gray marbled pattern on throat and chest (Fig. 3B); iris dark brown with golden speckles in the upper and lower thirds.

Coloration in preservative: After preservation in formalin and storage in ethanol, the general coloration pattern did not fade, dorsal coloration changed to darker grayish-brown, ventral surface of chest, belly, limbs turned whitish-beige; dorsal pattern, dark spots on flanks not changed, dark brown pattern changed to lighter brown; iris coloration faded and turned completely dark.

Measurements of holotype (all in mm): For comparative measurements see Table 2. Additional measurements: 2FL 2.7; 3FL 4.3; 4FL 2.4; 2TOEL 3.3; 3TOEL 5.0; 4TOEL 6.9; 5TOEL 3.1; N–SN 1.5.

Table 2 Measurements of the type series and referred materials on Micryletta nigromaculata sp. Nov.

No	Specimen ID	Type status	SVL	HL	SL	EL	N-EL	HW	IND	IOD	UEW	TYD	FLL	LAL	HAL	1FL	IPTL	OPTL	3FDD	HLL	
	Males		
1	ZMMU A5934	Holotype	22.7	6.9	2.7	2.9	1.9	8.1	2.2	3.3	1.8	1.2	16.3	12.3	7.0	2.0	0.6	0.8	0.5	34.7	
2	ZMMU A5935	Paratype	18.7	6.2	2.9	2.6	1.6	6.9	1.8	2.7	1.5	1.1	14.7	10.7	5.8	2.0	0.5	1.2	0.4	31.1	
3	ZMMU A5936	Paratype	18.8	5.9	2.6	2.5	1.6	6.7	1.8	2.5	1.5	0.9	13.3	9.8	5.3	1.8	0.5	0.7	0.4	30.7	
4	ZMMU A5937	Paratype	19.8	6.4	2.9	2.5	1.8	7.3	2.0	3.0	1.4	1.0	16.1	11.4	6.0	2.1	0.6	0.7	0.4	33.7	
5	ZMMU A5938	Paratype	20.4	6.7	2.9	2.7	1.8	7.4	1.9	2.9	1.5	1.2	16.8	12.1	6.2	2.1	0.6	0.8	0.4	35.4	
6	ZMMU A5939	Paratype	20.4	6.8	2.9	2.6	1.9	6.9	2.1	3.0	1.5	1.2	15.6	11.4	6.0	2.4	0.5	0.8	0.5	35.3	
7	SYS a007400	Paratype	20.4	6.3	2.9	2.5	1.9	7.0	1.9	2.9	1.5	1.0	16.1	11.3	5.8	2.2	0.6	0.7	0.5	35.5	
8	ZMMU A5940	Paratype	20.7	6.6	2.9	2.8	1.8	6.9	2.0	3.0	1.7	1.3	15.3	11.4	5.7	2.0	0.6	0.8	0.4	34.6	
9	ZMMU A5941	Paratype	20.7	6.6	2.9	2.8	1.8	7.2	2.0	3.0	1.6	1.1	17.1	11.9	6.1	2.1	0.6	0.8	0.5	35.6	
10	ZMMU A5942	Paratype	21.9	6.8	3.0	2.6	2.0	7.5	1.9	3.1	1.6	1.0	18.2	12.5	6.4	2.2	0.7	0.7	0.4	35.9	
11	ZMMU A5943	Paratype	22.0	6.7	2.9	2.6	1.8	7.4	2.2	3.2	1.5	1.1	17.7	13.1	6.9	2.8	0.6	0.8	0.5	39.0	
12	ZMMU A5944	Paratype	22.0	7.1	2.9	3.0	1.8	7.7	2.0	3.1	1.5	1.1	16.8	12.6	6.5	2.3	0.7	0.7	0.5	37.2	
13	ZMMU A5945	Paratype	22.3	6.9	3.1	2.8	1.9	7.1	2.0	3.1	1.6	1.2	17.2	12.7	6.7	2.4	0.5	0.8	0.5	38.2	
14	ZMMU A5946	Paratype	22.6	6.7	2.9	2.9	1.9	8.0	1.9	3.0	1.6	1.2	16.8	12.7	6.5	2.2	0.6	0.8	0.4	36.5	
15	ZMMU A5947	Paratype	22.8	7.1	3.0	2.6	1.8	7.5	1.9	3.0	1.6	1.2	17.8	13.0	7.1	2.5	0.6	0.9	0.5	39.0	
16	ZMMU A5948	Paratype	23.0	7.3	3.0	2.6	1.9	7.8	2.1	3.0	1.6	1.1	18.6	13.1	7.0	2.7	0.5	0.9	0.5	38.7	
17	DTU 301	–	18.5	5.7	2.4	2.5	1.6	6.1	1.9	2.6	1.4	1.0	13.3	9.6	5.3	2.1	0.5	0.8	0.4	31.3	
18	DTU 302	–	20.0	5.8	2.6	2.7	1.7	6.4	2.1	3.0	1.6	1.0	13.3	9.9	5.5	2.1	0.6	0.9	0.5	31.7	
		Mean	21.0	6.6	2.8	2.7	1.8	7.2	2.0	3.0	1.5	1.1	16.2	11.7	6.2	2.2	0.6	0.8	0.5	35.2	
		SD	1.5	0.5	0.2	0.1	0.1	0.5	0.1	0.2	0.1	0.1	1.6	1.1	0.6	0.2	0.1	0.1	0.0	2.7	
		Min	18.5	5.7	2.4	2.5	1.6	6.1	1.8	2.5	1.4	0.9	13.3	9.6	5.3	1.8	0.5	0.7	0.4	30.7	
		Max	23.0	7.3	3.1	3.0	2.0	8.1	2.2	3.3	1.8	1.3	18.6	13.1	7.1	2.8	0.7	1.2	0.5	39.0	
	Females		
19	DTU 303	–	24.2	7.1	3.1	2.9	1.9	7.3	2.1	3.3	1.7	1.1	16.7	12.0	6.6	2.6	0.7	0.7	0.6	36.7	
20	DTU 304	–	25.5	7.3	3.2	2.9	2.1	8.0	2.0	3.2	1.8	1.2	17.3	12.8	6.7	2.5	0.7	0.8	0.6	38.7	
21	DTU 305	–	25.9	7.0	2.9	3.0	2.7	7.5	2.3	3.3	1.8	1.2	16.1	11.9	6.2	2.4	0.7	0.7	0.6	35.6	
		Mean	25.2	7.1	3.1	2.9	2.2	7.6	2.1	3.3	1.8	1.2	16.7	12.3	6.5	2.5	0.7	0.7	0.6	37.0	
		SD	0.9	0.2	0.2	0.1	0.4	0.4	0.1	0.0	0.1	0.1	0.6	0.5	0.3	0.1	0.0	0.1	0.0	1.6	
		Min	24.2	7.0	2.9	2.9	1.9	7.3	2.0	3.2	1.7	1.1	16.1	11.9	6.2	2.4	0.7	0.7	0.6	35.6	
		Max	25.9	7.3	3.2	3.0	2.7	8.0	2.3	3.3	1.8	1.2	17.3	12.8	6.7	2.6	0.7	0.8	0.6	38.7	
No	Specimen ID	Type status	TL	FL	IMTL	1TOEL	4TDD	
	Males							
1	ZMMU A5934	Holotype	11.7	12.3	0.7	2.0	0.7	
2	ZMMU A5935	Paratype	9.9	9.0	0.7	1.8	0.5	
3	ZMMU A5936	Paratype	9.3	9.1	0.6	1.7	0.4	
4	ZMMU A5937	Paratype	10.7	10.8	0.6	2.1	0.4	
5	ZMMU A5938	Paratype	11.3	11.2	0.5	2.1	0.4	
6	ZMMU A5939	Paratype	10.7	11.2	0.6	2.2	0.5	
7	SYS a007400	Paratype	10.9	10.8	0.7	1.9	0.5	
8	ZMMU A5940	Paratype	11.1	10.9	0.6	1.8	0.4	
9	ZMMU A5941	Paratype	11.2	11.0	0.7	2.0	0.5	
10	ZMMU A5942	Paratype	11.3	12.0	0.6	2.1	0.5	
11	ZMMU A5943	Paratype	11.9	12.0	0.7	2.5	0.7	
12	ZMMU A5944	Paratype	11.6	11.6	0.7	2.2	0.5	
13	ZMMU A5945	Paratype	12.0	12.1	0.6	2.1	0.5	
14	ZMMU A5946	Paratype	11.9	11.3	0.6	1.9	0.5	
15	ZMMU A5947	Paratype	12.2	12.2	0.8	2.3	0.4	
16	ZMMU A5948	Paratype	12.2	12.4	0.7	2.3	0.5	
17	DTU 301	–	9.8	14.1	0.8	2.0	0.6	
18	DTU 302	–	9.6	13.5	0.9	1.9	0.6	
		Mean	11.1	11.5	0.7	2.0	0.5	
		SD	0.9	1.3	0.1	0.2	0.1	
		Min	9.3	9.0	0.5	1.7	0.4	
		Max	12.2	14.1	0.9	2.5	0.7	
	Females		
19	DTU 303	–	11.5	17.2	1.0	2.2	0.6	
20	DTU 304	–	12.3	18.4	1.1	2.4	0.8	
21	DTU 305	–	10.0	15.8	1.1	2.2	0.9	
		Mean	11.3	17.1	1.1	2.3	0.8	
		SD	1.1	1.3	0.1	0.1	0.2	
		Min	10.0	15.8	1.0	2.2	0.6	
		Max	12.3	18.4	1.1	2.4	0.9	
Note:

For character abbreviations see Materials and Methods.

Variation and sexual dimorphism. Individuals of the type series and the referred materials are generally quite similar in appearance and agree well with description of holotype, but show certain variation in coloration (Fig. 6). Dorsal color may vary from bright reddish-brown (Figs. 6A and 6B) to ochre-brown and light brown (Fig. 6C) and purplish brown (Fig. 6D). Dorsal pattern is very variable, in some specimens forming irregular confluent blotches, hourglass-shapes or “teddy-bear”-like shapes, see Rakotoarison et al. (2017) for definition, but are always edged with lighter (beige or orange) color. Size and position of black blotches on flanks also varies a lot, dark spots in sacral area may be connected (Figs. 4 and 6C) or disconnected (Figs. 6A and 6B) from the dark spot at groin, or may be absent in some females (Fig. 6D). Variation in size and body proportions of the type series and referred materials is given in Table 2. Females are larger than males: SVL 18.5–23.0 mm in males (N = 22) and 24.2–25.9 mm in females (N = 3). Females have comparatively larger body swollen with eggs, and comparatively shorter forelimbs (FLL/SVL mean ratio 77.4% (66.7–84.1%, N = 22) in males vs. 66.4% (62.4–68.9%, N = 3) in females). Males with single internal vocal sac. Skin texture appears to be less tuberculate in preservative than in life.

Figure 6 Color variation of Micryletta nigromaculata sp. nov. in life.

Cat Ba National Park: (A) Male paratype ZMMU A5945; (B) male paratype ZMMU A5935 in situ; Cuc Phuong National Park; (C) male DTU 302 in situ; (D) female DTU 303 in situ. Photos A–B by Nikolay A. Poyarkov; C–D by Tan Van Nguyen.

Distribution and biogeography: The presently known distribution of Micryletta nigromaculata sp. nov. is shown in Fig. 1. To date, the new species is known from limestone karst areas covered by primary tropical forest in Cat Ba N. P., Hai Phong Province, and by secondary tropical forest in Cuc Phuong N. P., Ninh Binh Province at elevations 90–150 m a.s.l. Northern Vietnam has one of the world largest areas of limestone landscapes, covered by specific limestone vegetation (Fenart et al., 1999; Day & Urich, 2000). The currently known range of Micryletta nigromaculata sp. nov. is divided by the vast lowlands of the Red River valley, an important biogeographic border in Indochina (Bain & Hurley, 2011; Yuan et al., 2016); our phylogenetic analysis estimates genetic divergence between the Cat Ba and Cuc Phuong populations at 0.7% (see Table 1). It is anticipated that Micryletta nigromaculata sp. nov. also occurs in the adjacent limestone karsts of northern Vietnam; in particular, records from Quang Ninh, Lang Son and Bac Giang provinces of northeastern Vietnam, as well as from Hoa Binh, Ha Nam and Thanh Hoa provinces of northwestern Vietnam are anticipated.

Natural history notes: Our knowledge on the biology of Micryletta nigromaculata sp. nov. is scarce; the species appears to be closely associated with karstic habitats. In Cat Ba N. P. (Hai Phong Province) during a 2-week survey in October 2011, specimens were only recorded from a small patch of limestone outcrops ca. 20 m in diameter, near a large limestone karst cliff and a small temporary body of water. Frogs were observed from 16:00 to 20:00 h hiding between small pieces of limestone rocks. Despite intensive search from 10 to 22 of October 2013, no additional specimens of the new species were recorded from other areas in Cat Ba N. P. In Cuc Phuong N. P. (Ninh Binh Province) specimens were found at night between 19:00 and 23:30 h near cave entrances and in valleys surrounded by limestone cliffs, relatively near to water sources. Surrounding habitat was limestone karst covered with primary polydominant tropical forest with multilayered canopy and an abundance of lianas, with occasional trees of Streblus macrophyllus (Moraceae), Terminalia myriocarpa (Combretaceae), Parashorea chinensis (Dipterocarpaceae) and Tetrameles nudiflora (Tetramelaceae) (in Cat Ba N.P.) or secondary forest (in Cuc Phuong N.P.). Reproduction biology, including advertisement call, tadpole morphology, as well as diet of the new species remains unknown.

Other species of anurans recorded syntopically with the new species at the type locality included Polypedates megacephalus Hallowell, P. mutus (Smith), Theloderma albopunctatum (Liu & Hu), Liuixalus calcarius Milto, Poyarkov, Orlov & Nguyen, Philautus catbaensis Milto, Poyarkov, Orlov & Nguyen, Hyla chinensis Günther, Microhyla butleri Boulenger, M. heymonsi Vogt and Micryletta cf. inornata. In Cuc Phuong National Park (Ninh Binh Province) the new species was recorded in sympatry with Occidozyga martensii (Peters), Leptobrachium guangxiense Fei, Mo, Ye & Jiang, Ophryophryne microstoma Boulenger, Glyphoglossus (formerly Calluella) guttulatus (Blyth), Microhyla heymonsi Vogt, Micryletta cf. inornata (Boulenger); Rana johnsi Smith; Sylvirana cf. annamitica Sheridan & Stuart; Raorchestes cf. menglaensis (Kou); Theloderma albopunctatum (Liu & Hu) and T. annae Nguyen, Pham, Nguyen, Ngo & Ziegler.

Genetic divergence. The new species is markedly distinct in mtDNA sequences from all congeners for which comparable sequences are available (mitochondrial gene 16S rRNA; uncorrected genetic distance ≥5.7%) and is reconstructed as a sister lineage with respect to all other examined members of the genus Micryletta.

Comparisons. Micryletta nigromaculata sp. nov. can be distinguished from all other congeners by external morphology and coloration, including presence of characteristic black patches on flanks and the hourglass-shaped irregular dark pattern on dorsum edged with thin orange line. From Micryletta erythropoda (Tarkhnishvili, 1994) (type locality in Dong Nai Province, known from lowlands of southern Vietnam) the new species can be distinguished by having generally smaller size in males (SVL 18.5–23.3 mm vs. up to 30 mm in M. erythropoda); by lacking outer metatarsal tubercle (vs. present in M. erythropoda); by having comparatively longer hindlimbs with tibiotarsal articulation of adpressed limb reaching the level of eye center (vs. reaching the level of the posterior edge of tympanum in M. erythropoda); by having dorsal surface slightly granular with small round flattened tubercles (vs. rather smooth dorsum in M. erythropoda); dorsum coloration brown to reddish-brown (vs. gray or beige to saturated ochre or brick-red in M. erythropoda); dorsum pattern with dark-brown irregular hourglass-shaped pattern edged with orange line and with two large black blotches in inguinal area (vs. extremely variable and formed by more or less dark contrasting spots on reddish background in M. erythropoda); lateral sides of head reddish brown without white patches (vs. dark brown with white spotting in M. erythropoda); flanks brown with dark patches or spots edged with white (vs. dark brown to gray with white patches in M. erythropoda); venter whitish with gray pattern (vs. brownish with violet tint in M. erythropoda).

Micryletta nigromaculata sp. nov. can be distinguished from M. inornata (Boulenger, 1890) (type locality in Deli, Sumatra; distributed through Malayan Peninsula to Myanmar, Indochina and southernmost China) by EL equal or shorter than SL (vs. snout shorter than the eye in M. inornata); IOD two times wider than UEW (vs. interorbital space just a little broader than the upper eyelid in M. inornata); dorsum coloration reddish-brown (vs. dark brown to violet in M. inornata); dorsum pattern with dark-brown irregular hourglass-shaped pattern edged with orange line and with two large dark spots in inguinal area (vs. more or less spotted or marbled with black blotches or longitudinal stripes in M. inornata); side of head dark-brown without white patches (vs. black with a series of white spots along the upper lip in M. inornata); flanks brown with dark patches or spots edged with white (vs. usually dark brown with white patches in M. inornata); venter whitish (vs. lower parts brown in M. inornata); throat in males whitish with light-gray marbling (vs. throat of males black in M. inornata).

Micryletta nigromaculata sp. nov. can be distinguished from M. steinegeri (Boulenger, 1909) (endemic to Taiwan) by having comparatively longer limbs with tibiotarsal articulation of adpressed limb reaching the level of eye center (vs. reaching the level of tympanum in M. steinegeri); dorsum coloration brownish to reddish-brown (vs. dark gray to violet in M. steinegeri); dorsum pattern with dark-brown irregular hourglass-shaped pattern edged with orange line and with two large dark spots in inguinal area (vs. inguinal dark spots absent, dorsum with irregular dark blotches or speckles in M. steinegeri); side of head uniform brown without white patches (vs. gray-brown with a series of white spots in M. steinegeri); body flanks with dark patches or spots edged with white (vs. flanks usually gray brown with dark marbling in M. steinegeri); venter whitish (vs. venter pinkish to orange in M. steinegeri).

Etymology: Specific epithet “nigromaculata” is an adjective in the nominative case, feminine gender, derived from Latin words “niger” for “black” and “maculatus” for “spotted,” in reference the characteristic black blotches on flanks in the new species.

Recommended vernacular names: We recommend “Black-spotted Paddy Frog” as the common English name of the new species and the common name in Vietnamese as “Nhái bầu hông đen.”

Conservation status: Micryletta nigromaculata sp. nov. is to date known only from two National Parks in northern Vietnam; in both localities frogs were recorded from very narrow specific limestone-associated habitats. It is important to notice that karst massifs in Vietnam, as well as in other parts of Southeast Asia, are facing ongoing severe threats from intensive deforestation and cement manufacturing; their continued exploitation for limestone cannot be stopped (Clements et al., 2006). This may be the major threat for the new species. Despite the actual distribution and population status of Micryletta nigromaculata sp. nov. are unknown, it is obvious that the new species is restricted to isolated highly endangered limestone karst massifs of northern Vietnam. It appears that the new species has strict habitat preferences since it was only recorded in a single locality in Cat Ba National Park (a small part of forest with limestone outcrops with an approximate diameter 100 m), and in a single habitat in Cuc Phuong National Park. Despite intensive searching efforts we could not record the new species in other surveyed areas of northern Vietnam. The two localities where the new species was recorded are isolated from each other and separated by a large area of unsuitable habitats—lowlands of the Red River and Gulf of Tonkin (see Fig. 1). At present, the Extent of Occurrence of the new species is estimated to be less than 100 km2, is severely fragmented and associated with rapidly declining limestone habitats. Additional surveys in other limestone areas of northern Vietnam are essential for elucidating the biology of the new species. Given the available information, we suggest Micryletta nigromaculata sp. nov. be considered as an Endangered (B1ab(iii), EN) species following IUCN’s Red List categories (IUCN, 2016).

Discussion

Our study provides an updated mtDNA genealogy and a new data on diversity of the genus Micryletta, which was not studied in detail in recent works on Microhylidae phylogenetics. The key study by Matsui et al. (2011) based on 12S–16S rRNA mtDNA fragment failed to recover phylogenetic placement of Micryletta within Microhylidae and concluded that this genus should be removed from the subfamily Microhylinae to form a distinct monotypic subfamily. These conclusions were not supported by consequent studies used multilocus phylogenetic approach, which all strongly suggested placement of Micryleta within Microhylinae radiation as a sister taxon to the group composed of Microhyla and Glyphoglossus (Peloso et al., 2016), or as a sister lineage of the clade joining Uperodon, Phrynella, Metaphrynella and Kaloula (Tu et al., 2018). Our study, though with moderate node support (0.94/78), places Micryletta as a sister lineage to the latter clade in agreement with results of Tu et al. (2018).

Matsui et al. (2011), based on analyses of three specimens of Micryletta, further showed that M. inornata was paraphyletic with respect to M. steinegeri, and argued that a sample of Micryletta sp. from Ranong Province in southern Thailand (which they identified as M. i. lineata) is more divergent than M. inornata from northern Thailand and M. steinegeri from Taiwan. Our study revealed a previously unknown species of Micryletta in northern Vietnam, which is proposed as a sister lineage with respect to all other examined populations (see Fig. 2). We also analyzed genealogical relationships between 17 samples of Micryletta from Vietnam, Laos, Thailand and Taiwan, including two topotype specimens of M. erythropoda from southern Vietnam and two populations of M. cf. inornata from northern Vietnam which are sympatric with the new species. Our data showed that M. erythropoda samples cluster with Micryletta sp. from Ranong Province and together they form a sister lineage with respect to all other populations of M. cf. inornata from Indochina and M. steinegeri from Taiwan. This lineage is clearly divergent from other M. inornata populations (4.6–5.9% in 16S rRNA gene; see Table 1) suggesting that M. erythropoda represents a distinct species, which occurs in lowlands of southern Vietnam and, possibly, also in southern Thailand. If identification by Matsui et al. (2011) is correct, the name M. inornata lineata (Taylor, 1962) should have the priority over M. erythropoda Tarkhnishvili, 1994. However, inclusion of topotype specimens of M. inornata lineata from Nakhon Si Thammarat Province of Thailand is required to revise this problem.

In our phylogeny (see Fig. 2) M. steinegeri from Taiwan was nested within the radiation of M. cf. inornata from mainland Indochina; the Taiwanese sample was only slightly divergent from M. cf. inornata (2.8% from M. cf. inornata group A, and only 1.3% from M. cf. inornata group B; see Table 1). Mainland populations of M. cf. inornata are grouped in several moderately divergent lineages, which also show significant variation in coloration (see Fig. 2), suggesting that taxonomy of this group might be incomplete. The revision of M. inornata–M. steinegeri group is currently not possible due to the lack of comparative materials from the type locality of M. inornata from Sumatra. However, all representatives of M. inornata–M. steinegeri complex, as well as M. erythropoda, can be easily diagnosed from the new species since they share several important diagnostic characters:

A certain degree of white spotting or the upper lip (see Fig. 2), whereas in M. nigromaculata upper lip is always dark brown lacking white markings (see Fig. 6). This character is also mentioned in the original description of M. inornata by Boulenger (1890) (“sides of head black, with a series of white spots along the upper lip”), reassuring us that the new species cannot be confused with M. inornata s.stricto.

Another important character is the relative length of snout, which is notably shorter than eye in all species of Micryletta, including M. inornata s.str. (Boulenger, 1890) but is subequal to EL in the new species (SL/EL 85.1–104.5%, mean 94.5%).

Throat in breeding males is dark (black to dark-gray) in all Micryletta species including the M. inornata s.stricto (Boulenger, 1890), but is whitish with gray marbling in M. nigromaculata.

Finally, the presence of dark inguinal spots was never reported for members of M. inornata–M. steinegeri complex, including the original description of M. inornata (Boulenger, 1890), but is characteristic for all examined specimens of M. nigromaculata.

Due to the wide range of M. inornata sensu lato (from northeast India through Myanmar to Indochina, Malay Peninsula and Sumatra) additional materials and further studies on many populations, especially from Sumatra, are critically required to solve taxonomic problems in this group.

Our study provides new evidence for previously unknown diversity of herpetofauna of karstic areas in Northern Vietnam. Previous studies in limestone massifs of Cat Ba Island in Ha Long Bay uncovered two new species of frogs (Milto et al., 2013) and one new species of gecko (Ziegler et al., 2008) all of which are endemic to the island and strongly associated with karst habitats. Cuc Phuong National Park and adjacent limestone massifs are also known for karst-associated endemism with a new species of gecko discovered from karst formations in this area (Ngo & Chan, 2011). Limestone karst massifs in northern Vietnam are divided by the Red River valley, an important biogeographic border in northern Indochina (Bain & Hurley, 2011; Geissler et al., 2015; Yuan et al., 2016). The discovery of M. nigromaculata population in Cuc Phuong National Park, on the other side of the Red River valley (see Fig. 1) provide further evidence for interconnection of limestone karst herpetofauna in northern Vietnam. Despite certain divergence in mtDNA 16S rRNA gene (0.7%), overall morphological similarity of the Cuc Phuong and Cat Ba populations of M. nigromaculata suggest they belong to a single species.

It is also remarkable, that in both localities M. nigromaculata was recorded in sympatry with M. cf. inornata, different from the new species both morphologically and genetically (see Fig. 2). The two-coexisting species of Micryletta however never were observed to share the same habitats, since M. nigromaculata was always restricted to very narrow patches of karstic limestone outcrops where M. cf. inornata was not observed.

Conclusions

Limestone karst areas are recognized as arks of highly endangered though still insufficiently studied biodiversity. Unique geological structure of karst massifs, formed by erosion and subterranean water drainages create numerous humid microrefugia with stable environmental conditions, which serve as an important environmental buffer for small vertebrates during periods of climate change (Clements et al., 2006; Glaw, Hoegg & Vences, 2006). The complex terrain of isolated karstic hills and caves create multiple ecological niches what along with their highly fragmented habitat-island nature result in high degrees of site-specific endemism within, and diversity among them (Oliver et al., 2017; Grismer et al., 2018). Limestone karsts are also known as important “biodiversity arks” for both surface and cave faunas, yet karstic regions are rapidly becoming some of the most imperiled ecosystems on the planet (Clements et al., 2006; Grismer et al., 2016a, 2016b, 2018; Luo et al., 2016; Suwannapoom et al., 2018). South-east Asia harbors more limestone karsts than anywhere else on earth (Day & Urich, 2000) with numerous new species including relic lineages of amphibians and reptiles being discovered from limestone areas (e.g. see discussions in Milto et al., 2013; Grismer et al., 2014; Grismer & Grismer, 2017; Grismer et al., 2016a, 2016b, 2017, 2018; Nazarov et al., 2014, 2018; Connette et al., 2017; Suwannapoom et al., 2018 and references therein). Ironically, though acting as major biodiversity hotspots, limestone karsts are critically endangered due to unregulated quarrying mostly for cement manufacturing, which is the primary threat to the survival of karst-associated species (Grismer et al., 2018); their continued exploitation for limestone cannot be stopped (Clements et al., 2006). Until karst habitats in Vietnam are thoroughly investigated, a significant portion of this country’s herpetological diversity will remain underestimated and unprotected. Our study thus calls for urgent focused survey and conservation efforts on karst herpetofauna in Southeast Asia and in Vietnam in particular.

Supplemental Information

Supplemental Information 1 Table S1. Specimens and 16S rRNA sequences of Micryletta and outgroup Microhylidae representatives used in molecular analyses.

GenBank AN–GenBank accession number.

Click here for additional data file.

We would like to thank the Department of Forestry, Ministry of Agriculture and Rural Development of Vietnam and the Forest Protection Departments of the Hai Phong and Ninh Binh provinces for supporting our fieldwork and issuing relevant permits. NAP thanks Andrei N. Kuznetsov (JVRTRTC, Vietnam), Leonid P. Korzoun (MSU, Russia) and Vyacheslav V. Rozhnov (IPEE RAS, Russia) and Nguyen Dang Hoi (JVRTRTC, Vietnam) for organizing and supporting his work in Vietnam. TVN thanks Thai Van Nguyen, Dung Van Le (SVW, Vietnam) for organizing and supporting his work in Vietnam. We thank Eduard A. Galoyan, Alina V. Alexandrova, Evgeniy S. Popov, Mikhail V. Kalyakin, Sergei V. Kruskop, Alexei V. Abarmov, Paul Freed, Olesya, Maxim and Stanislav Pavlovs’ (Russia) and Tan Nhat La (Hanoi) for help, support and encouragement during the fieldwork. We thank Le Xuan Son for help in the field and continuous support. NAP thanks Valentina D. Kretova for help with preparation of figures, Evgeniya N. Solovyeva for help with primer design, and Alexandra A. Elbakyan for help with accessing required literature. We are deeply grateful to Nikolai L. Orlov (ZISP, St. Petersburg, Russia) and Roman A. Nazarov (ZMMU, Moscow, Russia) for fruitful discussions. We are indebted to Valentina F. Orlova (ZMMU, Moscow, Russia), Natalia B. Ananjeva (ZISP, St. Petersburg, Russia) and Ying-Yong Wang (SYS, Guangzhou, China) for providing working facilities and an access to the collections under their care. We would like to express our gratitude to Marcio Pie, Mark D. Scherz and an anonymous reviewer for useful comments on the earlier version of this manuscript.

Additional Information and Declarations

Competing Interests

Author Contributions

Animal Ethics

Field Study Permissions

DNA Deposition

Data Availability

New Species Registration

The authors declare that they have no competing interests.

Nikolay A. Poyarkov conceived and designed the experiments, performed the experiments, analyzed the data, contributed reagents/materials/analysis tools, prepared figures and/or tables, authored or reviewed drafts of the paper, approved the final draft.

Tan Van Nguyen analyzed the data, contributed reagents/materials/analysis tools, prepared figures and/or tables, authored or reviewed drafts of the paper, approved the final draft.

Tang Van Duong conceived and designed the experiments, performed the experiments, analyzed the data, authored or reviewed drafts of the paper, approved the final draft.

Vladislav A. Gorin conceived and designed the experiments, performed the experiments, analyzed the data, authored or reviewed drafts of the paper, approved the final draft, logistic support.

Jian-Huan Yang analyzed the data, contributed reagents/materials/analysis tools, authored or reviewed drafts of the paper, approved the final draft.

The following information was supplied relating to ethical approvals (i.e., approving body and any reference numbers):

No experiments were conducted on living vertebrates. Specimen collection and animal use protocols were approved by the Institutional Ethical Committee of Animal Experimentation of Sun Yat-sen University (certificate number 2005DKA21403-JK issued to Ying-Yong Wang and Jian-Huan Yang).

The following information was supplied relating to field study approvals (i.e., approving body and any reference numbers):

Fieldwork, including collection of animals in the field, was authorized by the Department of Forestry, Ministry of Agriculture and Rural Development of Vietnam (permit number 1461/TCLN-BTTN, issued September 23, 2013).

The following information was supplied regarding the deposition of DNA sequences:

Sequences of 16S rRNA genes presented here are accessible via GenBank accession numbers MH756146–MH756156 and MH879840–MH879845.

The following information was supplied regarding data availability:

Specimens examined in this study are deposited in herpetological collections of the following museums: Zoological Museum of Moscow University (ZMMU, Moscow, Russia);

Duy Tan University (DTU, Da Nang Province, Vietnam);

The Museum of Biology, Sun Yat-sen University (SYS, Guangzhou, China).

The following information was supplied regarding the registration of a newly described species:

Publication LSID

urn:lsid:zoobank.org:pub:32150A60-5D04-4116-9816-0ED3E457504A.

Micryletta nigromaculata LSID

urn:lsid:zoobank.org:act:D7DA4B4C-04AD-4088-98A4-7EB1C18E6609.

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
