# Peer review of "A new limestone-dwelling species of Micryletta (Amphibia: Anura: Microhylidae) from northern Vietnam"

_PeerJ, doi:10.7717/peerj.5771_

## Round 0.1 · original submission · Minor Revisions

Both reviewers were enthusiastic about your study, but identified a number of issues that need to be corrected. Pay particular attention to the annotated file provided by Reviewer 1, as well as some important omissions indicated by Reviewer 2.

·

Basic reporting

This article is very well structured, and most of it is well written in good English; only in the introduction did I identify a number of grammatical errors, which I have taken the liberty of rectifying. The manuscript is well referenced and covers the breadth of literature on the topic of these frogs. The figures are nice, though a few adjustments are necessary, and the tables are also adequate, though at least one caption needs to be slightly adjusted. The authors should note that PeerJ does not expect references to be italicised, so they should check that their citation format fits that of the journal.

Experimental design

An exemplary taxonomic study, with apparent rigour and care taken. The description of poorly known taxa from threatened areas in southeast Asia (and indeed globally) is important. The authors should note some comments I have made on their methodology regarding measurements and repeatability for future studies, but these do not require action for the present study.

Validity of the findings

Findings appear robust, and the authors have clearly endeavoured to provide an accurate description of the new species and provide an exceptionally detailed diagnosis against other taxa that confirm its distinctiveness. The conclusions do a great job of framing the study in a conservation context. The authors should consider my comments on the conservation status of the new species, but need not change it if they prefer to leave the species as Data Deficient until further surveys can be conducted.

Additional comments

This paper is very well written and will require only minor revisions before it is acceptable for publication. I have made adjustments to the .docx file of the manuscript, and additional comments can be found there. I would like to see slightly more accurate colour terminology, and a few figures need to be slightly adjusted. Other changes and comments in the text are even more minor. In summary, I find this paper to be of high quality, and after these small revisions, I look forward to seeing it published in PeerJ.

Best regards,
Mark D. Scherz

Reviewer 2 ·

Basic reporting

Except for a few syntactical and stylistic errors, the language is easily comprehensible. Some of the figure captions need improvement. I suggest to the authors to scrutinize the MS several times to fix the minor errors.

Literature is adequately reviewed, but some deliberate omissions to important facts have been made, which I have explicitly pointed out below.

The data is shared and the article structure is adequate for a taxonomic description.

Experimental design

The authors take on a difficult taxonomic group and attempt to describe a new species, but there are several important points that are not adequetly covered, especially in regards to taxon sampling. I have highlighted these below in "General Comments to the Author".

The methods used are sufficiently described so that replication is possible.

Validity of the findings

The morphological description for the new species is impressive, and the within Vietnam sampling for a description of a new species is adequate. However, to become a significant contribution for the identification of the species unambiguously, more effort is needed, in the form of better taxon sampling and/or in deeper explanation.

Additional comments

The new species that you are describing is clearly a new species with both genetic and morphological differences. But you need to be sure fo the identity of M. inornata and M. steinegeri on your tree to give credence to the new species that is being described. M. inornata and M. steinegeri are not recognized individually on the tree. Since M. Inornata is the type for the genus, it should be identified unambiguously. For this, you would need a sample, ideally from Sumatra, Indonesia, the type locality for the species.

You say (57-59) “….Pyron & Wiens (2011) confirmed the validity of the genus
58 Micryletta and suggested that Micryletta is a sister taxon to the group composed of Microhyla, Glyphoglossus and Caluella, though with low values of node support….” But Pyron and Wiens also included Chaperina.

I also do not see Chaperina (C. fusca) on the tree (you have Glyphoglossus, Calluella, and Microhyla), which was suggested by some authors as a closely related sister taxon for Micryletta. Could your new species be a/the Chaperina? As the morphological features distinguishing these two are ambiguous and Chaperina has a large distribution from Malaysia, Thailand, Borneo to Philippines. Furthermore, Chaperina sequences from previous studies are available on GenBank.

This issue becomes important given that Micryletta as a genus is still poorly recognized.

Hence, I suggest that you show your genetic results on a tree which includes all the species, especially species which has voucher specimens.

Authors state (119-121) “….and has 120 been used in the most of recent phylogenetic studies on Microhylinae (Matsui et al., 2011; 121 Peloso et al., 2016; Nguyen et al., in press).” It is not advisable to use “in press” publications to support non-critical points. It is too obvious that it is for a self-citation, and is not a good practice.

(130) “…reducing 1 degree Celcius every cycle, and extension for 1 min at 72°C” consistently use “°C”.

(144) “For outgroups…” you have only one outgroup species. Outgroup is the lineage/s that you root the tree with.

Commas in front of “which” is absent in most instances. Many syntax related mistakes in the manuscript. Please scrutinize closely and correct.

Morphological comparisons with other Micryletta has been done through an analysis of the literature. Using the type specimen is the best way to do this without depending on descriptions of other authors. Since comparisons are based on literature, the clear demarcation and identification of the species on your tree becomes even more important (as I outlined above).

(401-402) “….Micryletta cf. inornata (Boulenger)..” So you have another Micryletta species co-occurign in the same habitat of the new species? This needs further explanation in the discussion. Has these been included in the tree? Or could this be variation of the new species?

(465-466) “…information, we suggest Micryletta nigromaculata sp. nov. be considered as a Data Deficient (DD) species following IUCN’s Red List categories (IUCN, 2016).” Having provided distribution, genetic, morphological and natural history data, one should not claim that species to be data deficient. Then it also looks as if you are undermining your own efforts. There are so many species which do not have even basic distribution data and these are the ones that are generally referred to as data deficient by the IUCN. Hence, try to evaluate the species at least based on Area o f Occupancy and Extent of Occurrence.

Figure 3 Caption “Male holotype of Micryletta nigromaculata sp. nov. (ZMMU A5934) in life”. Awkward phrasing. You could say “Holotype of Micryletta nigromaculata sp. nov. (ZMMU A5934), male, in life”. But some of the pictures suggest that the animal is not photographed “in life”. Please check.

Figure 4 caption - It is awkward to say “male holotype…” you could say “Holotype of M. nigromaculata sp. nov. (ZMMU A5934), male….”

---

## Round 0.2 · accepted · Accept

I think that, given the limitations of access to additional tissue samples, you were able to address all of the issues raised by the reviewers.


#